# Analysis of Home Healthcare Practice to Improve Service Quality: Case Study of Megacity Istanbul

**DOI:** 10.3390/healthcare11030319

**Published:** 2023-01-20

**Authors:** Rabia Çevik İnaç, İsmail Ekmekçi

**Affiliations:** 1Department of Industrial Engineering (Ph.D. Program), Institute of Pure and Applied Science, Istanbul Commerce University, Kucukyali, Istanbul 34445, Turkey; 2Department of Industrial Engineering, Istanbul Commerce University, Kucukyali, Istanbul 34445, Turkey

**Keywords:** home healthcare services, regression model, machine learning algorithms, estimation, performance measurements, service quality, Istanbul

## Abstract

Home healthcare services are public or private service that aims to provide health services at home to socially disadvantaged, sick, needy, disabled, and elderly individuals. This study aims to increase the quality of home healthcare practice by analyzing the factors affecting it. In Megacity Istanbul, data from 1707 patients were used by considering 14 different input variables affecting home healthcare practice. The demographic, geographic, and living conditions of patients and healthcare professionals who take an active role in home healthcare practice constituted the central theme of the input parameters of this study. The regression method was used to look at the factors that affect the length of time a patient needs home healthcare, which is the study’s output variable. This article provides short planning times and flexible solutions for home healthcare practice by showing how to avoid planning patient healthcare applications by hand using methods that were developed for home health services. In addition, in this research, the AB, RF, GB, and NN algorithms, which are among the machine learning algorithms, were developed using patient and personnel data with known input parameters to make home healthcare application planning correct. These algorithms’ accuracy and error margins were calculated, and the algorithms’ results were compared. For the prediction data, the AB model showed the best performance, and the R^2^ value of this algorithm was computed as 0.903. The margins of error for this algorithm were found to be 0.136, 0.018, and 0.043 for the RMSE, MSE, and MAE, respectively. This article provides short planning times and flexible solutions in home healthcare practice by avoiding manual patient healthcare application planning with the methods developed in the context of home health services.

## 1. Introduction

Home healthcare services are provided to individuals who need them due to several diseases, including social and mental counselling services in their homes and families [1]. Home care services try to lessen the effects of illness and disability by making it easier for people to live their daily lives, finding the best way to treat them, and improving their quality of life [2]. This service is also suitable for those who prefer to stay at home and whose treatment and care continue, but it is necessary for those who cannot be cared for by their close family and friends.

Home health services include various health services offered in their own homes upon the requests of patients who cannot access health institutions. Home health services are attractive because of their advantages when the treatment is carried out in a hospital or home environment [3]. Patients, healthcare workers, and health institutions are the three most important factors in the home health services process [4]. The problem becomes quite complex when the specific constraints of all stakeholders are brought together. Controls in health institutions include how long it takes to plan, how often decisions are made, how long care stays the same, where it works, and what kind of service it offers [5,6]. Concerning the patient, things such as how often they visit, how long they wait between visits, how well the services work together, and their own personal preferences are considered. From the point of view of health personnel, the type of contract, working hours, workload, skills, and personnel position limit the problem [7].

Home Health Service is meeting the health, care, and rehabilitation requirements of patients, individuals needing care, and their relatives in their home environment [8]. Home care is provided during the diagnosis and post-treatment care process of a sickness or disorder, as part of the follow-up of a chronic illness, or as part of preventive health and examination services. Home care service is offered to the disabled, elderly, and people in need of care in their environment [9]. Home care service aims to minimize the effects of the disease and disability by applying the most proper treatment while affecting the least daily living conditions and, at the same time, increasing the value of the patient’s life. Health services are provided at home or in health centers similar to the home environment to boost patient morale and motivation worldwide [10]. The most important benefit of home health services is that the person feels at peace in the environment and is cared for when he is with his family [11]. For this reason, home care services should be provided to all disadvantaged groups (elderly, sick, bedridden, disabled, etc.,), and existing care centers should be accessible to all patient groups. The following purposes will be achieved with home health services:Supporting the target group in terms of health and social aspects in the home environment,Continuation of the recovery period without interruption,Accelerating the healing process,Reducing hospital infections,Reducing the length of hospitalization and the stress of hospitalization,Lowering the cost of health,Supporting the traditional family structure,Making the patients and their relatives educated and self-sufficient and creating a suitable environment where the patients live comfortably with their health conditions.

The increase in the aging population worldwide, the development of medical technology, and economic factors have led to a severe rise in home health services. Due to high demand, healthcare organizations have had to deal with several problems at both the strategic and operational levels [12]. Researchers are interested in the problems that health institutions face when trying to give good service. Artificial intelligence (AI) and machine learning (ML) models are widely used by researchers in the health field, both in terms of medicine and management, to solve problems that come up in the health field. With the developed methods and practices, the main goal in the field of health is to ensure that decisions are taken correctly, which will effectively reach the desired quality level [13].

Among the abilities classified as AI today, it includes the successful understanding of human speech, high competitiveness in self-driving cars in strategic game systems, intelligent directing in contented transfer networks, armed simulations, and understanding of multifaceted data [14]. AI-based algorithms will be used to reorganize care, relieve home healthcare workers of complicated, monotonous tasks, improve healthcare for a larger population at lower costs, and, in the end, make adaptive predictive, preventive, and participatory care easier [15]. However, AI solutions have not extensively entered typical home healthcare applications, mainly due to (1) restricted data convenience, approachability, structure, and exhaustiveness; (2) lack of procedural accuracy and ideals in their advance; (3) and applied queries around the value and practicality of the solutions, but also ethics and accountability [16]. With recent advancements in computer technology, machine learning (ML), a sub-branch of AI, is being used in many fields, including healthcare.

ML is a method in which algorithms are used to learn the fundamental statistical patterns and structures in data and to figure out the predictive values of data that have not yet been seen. An ML model outperforms complex data structures such as imagery or language [17]. Today, ML comprises many disciplines, such as statistics, information concepts, algorithm theory, possibility, and functional investigation [18]. ML algorithms have supervised (dependent variable having values according to independent variables) and unsupervised learning types. This study considered the supervised learning type since the dependent variable has a value compared to the independent variables. ML algorithms with supervised learning types generally contain variables with the categorical data type [19]. Typically, regression-based ML algorithm performs well in supervised learning type. In this study, regression techniques, which form the basis of ML algorithms, are integrated with ML models.

One of the most important aims of this study is to estimate the required home healthcare time for a patient with ML algorithms. Although the home health service application depends on many factors, such as the patient’s health, age, gender, and disability, it usually spreads over a long period. A study of long-term home care recipients in the UK emphasized that quality of care has many dimensions, including safety, experience, and effectiveness. The interdimensional relationship is essential for policy and practice [20]. Another study developed a model formulation based on variable neighborhood search and a metaheuristic solution approach to optimize nurses’ daily schedules for primary healthcare providers employed in Austria, yielding results that will affect home health service duration [21]. In this study, fourteen factors were discussed under the main parameters, such as the independent variables that are thought to be effective on the dependent variable, such as demographic, geographic, and living conditions of the employees and patients. Besides analyzing the effects of these factors, their influential role in estimating the time required for a patient was examined.

The regression method was used to analyze the strength and direction of the effects on the duration of home healthcare practice [22,23,24]. Researchers widely use this method, especially in health management [25]. In a study, the nonlinear regression method planned health resources working in a health institution, reducing long waiting times and increasing patient satisfaction [26]. Another study used linear and logistic regression models to determine what makes a hospital more likely to be in the bottom 25% of all hospitals reported [27]. Vainieri et al. used the regression method to look at the relationship between senior management skills, knowledge sharing, and organizational performance in the public health system and to ensure that professionals were involved [28]. Another study used a segmental regression analysis to determine how long patients stayed in the hospital after the long-term care insurance (LTCI) program started [29]. Most of the time, regression analyses are used to measure how independent variables affect dependent variables and show how they affect each other [30]. Moreover, regression models let you estimate the value of the dependent variable by figuring out the values of the term coefficients of the independent variables. But to check if the results of the regression model analysis are correct, the low R^2^ values cause the estimation data to be inconsistent. In this study, regression-based ML algorithms were used to estimate how long a patient can receive home healthcare on average.

This study consists of four main parts. The first part of the study discusses the literature review, revealing the importance and necessity of home healthcare practice and evaluating the affecting factors. The remainder of the article is structured as follows: Detailed information about the regression method and ML algorithms for the recognition and processing of accurate data used in this study is given in the Section 2. The statistical and estimation results of the study are discussed in the Section 3, and the effects of the numerical results obtained by the proposed method are also discussed. The technique, results, and interpretations used in this study are discussed in the last part to contribute to other studies.

## 2. Methodology

In this study, linear regression and machine learning algorithms were run using data from a home health service application in Megacity Istanbul. The characteristics of the data used in the study and the theoretical explanations of the methodology are discussed in detail in the continuation of this section.

### 2.1. Data Compilation

Actual data on home healthcare practice in Istanbul were analyzed for this study. In the study, the data of the home health services provided to approximately 13,806 patients were passed in the cleaning process, and all data of 1708 patients were analyzed. Istanbul was chosen as the plot region for the home health service application, which continues in approximately 37 areas of this city. Istanbul is the city with the highest population density in Turkey. For this reason, the home health service application data used in this city constitutes a potential example for other regions. In this study, the data set of the study was created by considering 14 independent variables and one dependent variable type.

The chosen dependent variable in this study was “Service Period (Year) or (Day)” which means service lifecycle. The aim of the chosen this dependent variable was to ensure the sustainability of the home care service, it is necessary to analyze the service lifecycle and improve it in this regard. This is an essential issue for quality improvements in home health systems. By knowing the time elapsed until the recovery period of the patients who request home health care services, the work plans of the health personnel employed in a limited number are made, allowing a time frame to be formed for the patients waiting for home health care service. This service, which is offered with limited resources, should be carried out with a sufficient number of health professionals. Service providers (Public and Private Sector) need to know how long it takes to serve the new patient who will join the system and what are the factors affecting this time.

Table 1 contains the definitions, notations, and data types of the dependent and independent variables preferred for this study.

The input variable “AptNo” represents the number of the location where a patient is receiving home health service lives. With this variable, information about the ease or difficulty of transportation of the personnel providing home health services is obtained using the floor information where a patient’s house is located. Since the people living in detached or single houses (generally single-floored houses) did not have floor information, these patients’ house numbers were considered 0.

The districts where the homes of the patients who benefit from home health services are located were evaluated as the location parameter. This study contains information about 37 districts and four central regions to which these districts are affiliated. The city of Istanbul has two separate sides, the Anatolian and European sides, with the Bosporus it owns. On each side, two regions are responsible for home health service implementation. Two of these regions (Edirnekapi and Kucukcekmece) are responsible for the districts on the European side, and two (Kartal, Uskudar) are responsible for the Anatolian ones. The demographic structures of the patients were analyzed by considering patient age and gender as input parameters. The same feature has been studied regarding the quality of home health service applications, considering the age, gender, and seniority of the health personnel who perform the home health service application.

Independent variables usually have categorical data types, including patients’ medical and living conditions. A patient has one or more chronic diseases. Home health services were examined in detail using data on whether these patients lived alone, with a family member, or with a companion. In addition, the disability status of patients receiving home health services, such as bedridden, vision, hearing, and mental disabilities, were considered. The study’s independent variables have demographic data types such as age and gender, mainly belonging to patients and personnel serving patients. Descriptive statistics of input and output factors considered in this study are given in Table 2.

The histogram plot of the dependent variable showing the skewed data is shown in Figure 1. The daily data of the independent variable were analyzed as right skewed (skewness value 3.46). The histogram with data skewed to the right shows the time required for home healthcare practice for a patient. Most wait times are relatively short (under 90 days, which gives the median) and only enter the third quartile data, and home healthcare time is extended. The data entering the first quarter of the data set were analyzed to be less than 38 days.

### 2.2. Methodology of the Study

The flowchart that visualizes the working of the methodology proposed in this study is displayed in Figure 2 to analyze the dependent and independent data within the methods of this study. This study analyzed dependent and independent data using two methods to obtain numerical results. To measure the effects of the independent variables, which is one of the methods, on the dependent variable, the linear regression method, including the ANOVA analysis, was used.

Traditional linear or nonlinear regression analyses are run to measure the effects of independent variables on dependent variables. It was used to analyze the impact of 14 independent variables in this study on the duration of home healthcare practice required for a patient. The effect coefficients and statistical significance levels of independent variables on dependent variables were calculated. In a linear regression model, the effect coefficients of the independent variables and the positive or negative aspects of the coefficients constitute the regression equation. Statistical methods lie based on the ML algorithms used in the second stage of the methodology part of this study. For this reason, this study evaluated dependent and independent variable data by integrating traditional statistical methods and machine learning algorithms. Estimation data based on linear regression analysis is provided through an equation obtained by considering the effects of independent variables on the dependent variable. ML algorithms discover common aspects by training and reading all the data in the data set and linking the prediction data with the actual data to obtain the prediction data of ML algorithms. Most of the ML algorithms are based on the mathematical formation of the linear regression method. For this reason, the estimation data of ML algorithms have a higher accuracy rate and less error margin than the estimation data of linear regression.

The duration of the home health service application needed for a patient was estimated using ML algorithms in the second stage of the study. Among the ML algorithms, random forest (RF), adaptive boosting-AdaBoost (AB), gradient boosting (GB), and neural network (NN) are the most utilized by researchers and were preferred in the present research. Each ML algorithms efforts to forecast the response variable with different models. The best results of estimation values were attained in the present research using the algorithm’s feature selection choice. The following equations are created for feature selection [31,32]:(1)$$F\left(g_{i},\;T\right)=\int p\left(g_{i},\;T\right)\ln(\frac{p\left(v_{i},\;T\right)}{p\left(v_{i},\;T\right)}{\mathrm{dg}}_{i}\mathrm{dT}$$
(2)$${\mathrm{oT}}_{f}=\frac{1}{\left|\varphi \right|}\;\underset{v_{i}\in \varphi }{\sum }F\left(v_{i},\;T\right)-\frac{1}{{\left|E\right|}^{2}}\;\underset{v_{i}v_{\mathrm{ij}}\in \varphi }{\sum }F\left(v_{i},{\;v}_{j}\right)$$
where φ represents the duration of the home health service, and F symbolizes the feature selection information. $v_{i}$ signifies the time of the home health service. T denotes the independent factor variety. $F\left(v_{i},\;v_{j}\right)$ signifies the collaborative information between $v_{i}$ and $v_{j}$. $oT_{f}$ symbolizes the number and type of independent factors to regulate the feasible feature selection. Virtually 1536 data of all data (90%) were run for the training phase, while the rest of the data (10%) were used for the testing phase [33]. Orange 3.33 version was used to obtain predicted results of ML algorithms in this study. The software screenshot of AB, RF, NN, and GB models from ML algorithms is shown in Figure 3.

### 2.3. Machine Learning Algorithms

In this study, four different ML models were operated to predict the duration of home healthcare performed at the request of a patient. Detailed information about these algorithms is discussed in the following subsections of this section.

#### 2.3.1. Random Forest (RF)

RF algorithm with supervised learning feature is frequently preferred in solving regression and classification problems [34]. The basis of the RF algorithm is that each leaf node corresponds to a class label and the attributes determined in the algorithm represent the inner node of the trees [35]. Decision trees, which enable the output variable in RF models to take values continuously, are also known as regression trees. For this reason, in this study, by integrating the RF algorithm with linear regression analysis, the processes of determining and estimating the independent variables that are effective on the output variable are carried out.

The attribute information of the RF algorithm used in this study is as follows: Number of trees, 10; Maximal number of considered features, unlimited; Replicable training, No; Maximal tree depth, unlimited; Stop splitting nodes with maximum instances, 5.

Especially for a data set with a categorical data type, the RF algorithm performs very well. The assumptions of the RF algorithm in this study are listed as follows:The data in the training data set was accepted as the root.Since the data types in the data set are categorical, the discretization of continuous data is unnecessary.The records of the RF algorithm are distributed recursively.Statistical methods are used to rank the roots and nodes of this algorithm.Pruning is based on the RF algorithm to avoid overfitting.

#### 2.3.2. AdaBoost (AB)

The boosting method creates a strong learner by bringing together many weak learners in ML algorithms [36]. The most common algorithms using this method are the AB and GB models. The basis of these two algorithms is to train the predictive variables by making them sequential [37]. The AdaBoost algorithm is also known as Adaptive Boosting. This algorithm produces a ranking by determining the priority relationships among the estimators [38]. The AB algorithm draws a path to smooth out the training data that the preceding estimator learned poorly. The attribute information of the AB algorithm used in this study are as follows: Base estimator, tree; Number of estimators, 50; Algorithm (classification), Sammer; Loss (regression), Linear.

#### 2.3.3. Gradient Boosting (GB)

The GB algorithm is a supervised ML algorithm for regression and classification samples [39]. This algorithm provides an estimation model in the form of a collection of weak estimation models with decision trees such as the AB algorithm. Based on this algorithm, regression trees, which are accepted as the primary learner, are based on a serial estimator tree based on the errors calculated by the previous tree. The attribute information of the GB algorithm used in this study is as follows: Method, Gradient Boosting (scikit-learn); Number of trees, 100; Learning rate, 0.1; Replicable training, yes; Maximum tree depth, 3; Fraction of training instances, 1; Stop splitting nodes with maximum instances, 2.

#### 2.3.4. Neural Network (NN)

The basis of NN algorithms lies in obtaining predictive data using synaptic connection strengths between neurons that represent simple processing units [40]. An NN algorithm acts as a massively parallel distributed processor composed of artificial neurons [41]. In this algorithm, the connections between the data are revealed by the signals between the neurons, and the prediction data are formed. Signals formed between neurons move from the first layer to the last layer by switching between the hidden layers [42]. These signals can pass between layers more than once. In this study, 100 hidden layers were used for the NN algorithm. In addition, a ReLu function is defined for this algorithm that takes the weighted sum of all the inputs in the previous layer, generates a predictive value, and passes it to the next layer [43]. The attribute information of the NN algorithm used in this study is as follows: Hidden layers, 100; Activation, ReLu; Solver, Adam; Alpha, 0.0001; Max iterations, 200; Replicable training, True.

### 2.4. Evaluation Parameters

Estimating the home health service duration was provided using six different ML algorithms in the present study. The performance measurements require to be computed to compare these algorithms. These criteria are included in ML algorithms as the coefficient of correlation (R^2^), mean absolute error (MAE), root mean square error (RMSE), and mean squared error (MSE), respectively. The equations used for performance measurement are given below [44]:(3)$$\in =y_{a}-{\tilde{y}}_{a}$$
(4)$$\mathrm{MSE}=\frac{1}{n}\overset{n}{\underset{a=1}{\sum }}{\left(y_{a}-{\tilde{y}}_{a}\right)}^{2}$$
(5)$$\mathrm{MAE}=\frac{1}{n}\overset{n}{\underset{a=1}{\sum }}\left|y_{a}-{\tilde{y}}_{a}\right|$$
(6)$$\mathrm{RMSE}=\sqrt{\frac{\sum_{a=1}^{n}{\left(y_{a}-{\tilde{y}}_{a}\right)}^{2}}{n}}$$
(7)$$R^{2}=\overset{n}{\underset{a=1}{\sum }}{\left[\frac{{\tilde{y}}_{a}-y_{a}}{y_{a}-{\bar{y}}_{a}}\right]}^{2}$$
where $y_{a}$, ${\bar{y}}_{a}$, and ${\tilde{y}}_{a}$ signify the real value in the data set, the mean value of the data set, and the forecast values produced by ML algorithms. n is the number of observed data (sample) in a dataset. The value of the margin of error ∈ calculates the actual and predicted data. RMSE, MAE, and MSE are performance measurement tools that decrease predictive values in ML algorithms. The value of R^2^ takes a number between 0.00 and 1.00. The selection of determining the factors that will give the feasible result for which algorithm by applying feature selection is widely used in the algorithms of ML.

## 3. Results

### 3.1. Statistical Analysis Results of Independent Factors

In this study, the linear regression technique was used to obtain numerical results of the statistical model of dependent and independent variables. Table 3 shows the statistical results of the independent factors that affect the home health service application duration needed for a patient. Term coefficient, standard deviation coefficient, t-value, statistical significance *p*-value, and VIF data were calculated to determine the effect power and direction of the input factors on the output variable. These data were shared in Table A1 in Appendix A of the study.

As a result of the statistical regression analysis, 14 independent variables were analyzed, and two independent variables had no or minimal effect on the dependent variable. It has been statistically evaluated that the health personnel who comes for a patient’s home health service is not affected by the number of floors of the apartment where the patient lives. The statistical *p*-value of the AptNo independent variable was calculated as 0.129. The statistical *p*-value of the other independent variable with little or no effect was calculated as 0.526. This variable was included in this study as an answer to whether a companion accompanies a patient. The presence of a person with a patient throughout the day has little effect on the duration of home healthcare. The standardized effect graph of the parameters that directly or indirectly affect the duration of the home health service application needed for a patient is shown in Figure 4.

A Pareto chart (standardized effect) was preferred to determine the magnitude and importance of the impact of independent variables. Bars crossing the baseline (baseline value calculated as 2 for this study) in this graph are considered statistically significant. Bars representing all independent factors except A (AptNo) and K (Wh/Woh) independent factors in this study cross the reference line at 1.96. These independent variables are statistically crucial at the 0.05 level in the statistical regression model and are effective on the dependent variable.

### 3.2. ML Algorithms Results of Dependent Factors

The data set was analyzed in two stages, training and testing, and the prediction and accurate data were compared in RF, AB, GB, and NN models from ML algorithms. For the training and testing phases, the data set was shared at a rate of 90%/10% for ML algorithms. There is a total of 1707 data in the data set to run in the ML algorithms. ML algorithms were run for 170 of these data for testing and the remaining data for the training phase. The training and test data are entirely random, and the selected data are shown in Figure 5.

The performance measurement values of the RF, AB, GB, and NN algorithms of the data used for the training and testing phases required for ML algorithms are given in Table 4. The AB algorithm has the highest R^2^ values in the training and testing phases. R^2^ values of 0.990 and 0.903, respectively, were obtained for these phases of the AB algorithm. While the GB algorithm has the lowest R^2^ value in the training phase, the NN algorithm has the lowest R^2^ value in the testing phase. The RF model, the second-best algorithm after the AB algorithm, has high R^2^ and inferior margins of error.

The performance measurement results of the algorithms used to verify the validity of the estimation data of the output variable of the ML algorithms are analyzed. The performance measurement results of the ML models utilized in the present research are given in Table 5. Correlation values of ML algorithms have been measured since the correlation values, one of the performance criteria, express the connection between the estimated results obtained by the algorithms and the actual results. As a result of statistical data of other algorithms, the coefficient of correlation of RF, AB, GB, and NN values was calculated as 0.861, 0.903, 0.733, and 0.521, respectively.

The RSME values of RF, AB, GB, and NN were calculated as 0.163, 0.136, 0.225, and 0.302, respectively. The algorithms of RF and AB have fewer errors than the RMSE values of the algorithms. The MSE values of RF, AB, GB, and NN were estimated as 0.026, 0.018, 0.051, and 0.091, respectively. ML algorithms of RF and AB have data of consistent estimation results with low MSE values. The MA values of RF, AB, GB, and NN were computed as 0.078, 0.043, 0.175, and 0.164, respectively. RF and AB algorithms have lower SE values than other ML algorithms in this performance criterion and the values of RMSE and AE. According to the performance measurement criteria values, the AB algorithm provides the best prediction, while the NN algorithm provides the worst prediction data for the response variable.

The comparison of the estimation data of the ML algorithms with the actual data is shared in Figure 6. Among the algorithms, the AB model gave the best prediction results. It is detected that there is less difference between the data of the AB algorithm and the actual data in the case of keeping the closeness of the estimated data and the real data in the figures. There is no overlap between the prediction data of other algorithms and the actual data. However, it has been observed that the exact number of data is low because the prediction data of the EU algorithm overlaps with the actual data.

The average value of the actual data used for the test phase of the home health service application data set was calculated as 164 days. According to the whole data set, the average home health service period required for a patient is 200 days. The mean values of the prediction data of the RF, AB, GB, and NN algorithms were calculated as 169, 164, 169, and 178, respectively. According to the test data set, the AB algorithm’s average value and the actual data’s average value overlap. The mean prediction time of the RF, NN, and GB algorithms deviated by 2.96%, 7.87%, and 2.96% from the actual times, respectively. The standard deviation value of the data selected for the actual test was calculated as 100 for the daily and 0.25 for the year. The daily standard deviation values of the RF, AB, GB, and NN algorithms for the home health service duration were calculated as 83.67, 89.76, 50.42, and 95.99, respectively. The annual standard deviation values of the home health service duration of these algorithms were computed as 0.28, 0.23, 0.25, 0.14, and 0.26, respectively.

The prediction data of the RF, AB, GB, and NN algorithms are compared in Figure 7. Among the results obtained for 170 test data selected from the actual data, the estimation results of the AB algorithm are closer to the actual data. On the other hand, the RF algorithm exhibits the second-best performance after the AB algorithm with its high R^2^ value and low margins of error, equaling the high level of consistency in the prediction data. Running the developed ML algorithms allows one to calculate the home health service time for a patient. Thus, this study provided the opportunity to estimate the required duration of home service according to the demographic, geographical, and life-related parameters of a patient demanding home health service. In addition, some factors that will affect the duration of home health service play an essential role in the planning of home health service application according to the characteristics of the input parameters of the health workers who play an active role in the home health service application. This study provides the opportunity to provide quality service in the home health service application by making a significant contribution to the estimated duration of home health service, both in the number of personnel and in the patient admission conditions.

This study has some limitations. The first limitation is the literacy rate data was not selected among the independent parameters to reflect the education level in the country. Otherwise, this study does not directly include the education level factor when there is no long-term data on education level. Another limitation is that the home health workforce is limited only to those working in government-supported institutions. Since there is no official data on the home health workforce in private home care institutions, they were not included in the study. The last limitation of this study is that other factors, such as disease and smoking rates in the plot city, were not considered to calculate the estimation data.

Health expenditure rates in countries are increasing day by day. One of the most significant shares in health expenditures is healthcare resources employed [45]. Using alternative resource allocation techniques in home healthcare, more information about the economic impact can be provided to decision-makers, namely policymakers [46]. Inefficient resource management negatively affects health systems in terms of cost and time. For this reason, it is inevitable to use the ML method, one of today’s most up-to-date methods. This study offers a solution to the resource management problem in health management with ML algorithms.

## 4. Conclusions

The increase in the elderly population worldwide, the development of medical technology, and economic factors have led to a severe rise in home health services. As a result of the demographic trends of people day by day, there will be an increase in demand for home health services in the future. As a result of the rise in the elderly population, especially in parallel with the formation of long-life spans, as well as the increase in diseases and the long struggle of people with these diseases, today’s health systems have to turn to home health services.

Home health service is an essential application for health systems to continue examination, treatment, physical therapy, cleaning, and follow-up procedures for patients who have limited access to any health institution due to various health problems and are therefore deemed appropriate to receive health care services at home. The patient group, which is one of the most critical stakeholders of home healthcare, is listed below, and this patient group will benefit significantly:Patients who need wound care, need a catheter, continue their treatment by injection, and who will be given serum by vascular access,Patients with chronic disease,Terminal patients,Patients who have been released from health institutions and need to continue their treatment at home,Those who need physiotherapy and psychotherapy at home.

ML algorithms provide the opportunity to know the problems that may occur by obtaining predictive data and sensitivity analyses in many areas. The potential of ML in overcoming the complexity of various data and the difficulties encountered in home healthcare management has been revealed in this study. ML algorithms are generally used in medical diagnosis in the field of healthcare. This paper showed using algorithms that allow ML to be developed for predictive data to enable more data-driven health informatics and management solutions. By running ML algorithms, namely, RF, AB, GB, and NN, performance measurement criteria values were compared to show the consistency of the estimated data on the duration of the home health service application. Among ML algorithms, the algorithm with the best performance was defined as the AB algorithm, while the model with the worst prediction data was determined as the NN algorithm. The AB algorithm has a high correlation value of 0.903, and the RMSE, MSE, and MAE values representing the margins of error of this algorithm were calculated as 0.136, 0.018, and 0.043, respectively.

The implications for healthcare research are that there is no doubt that health system management will become more dependent on ML algorithms of complex data such as healthcare resources, health economics, patient rights, patient and health professional satisfaction, and so on. Finally, we present an example of future applications for ML algorithms that will significantly impact the healthcare system management. This study aimed to help the future planning of a country in terms of home healthcare service by calculating the estimation data of the duration of the home health service by ML algorithms.

Home health services are a service that must be maintained using limited resources. Although technological advances and telehealth services facilitate the services of health professionals, estimating the completion of the service cycle is a crucial issue for resource planning. This study provides service providers with a predictive model for the completion of the service cycle.

## Figures and Tables

**Figure 1 healthcare-11-00319-f001:**
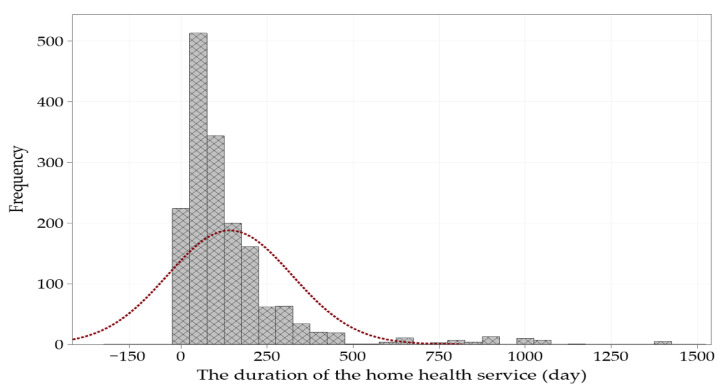
Histogram with fit lines of the dependent variable.

**Figure 2 healthcare-11-00319-f002:**
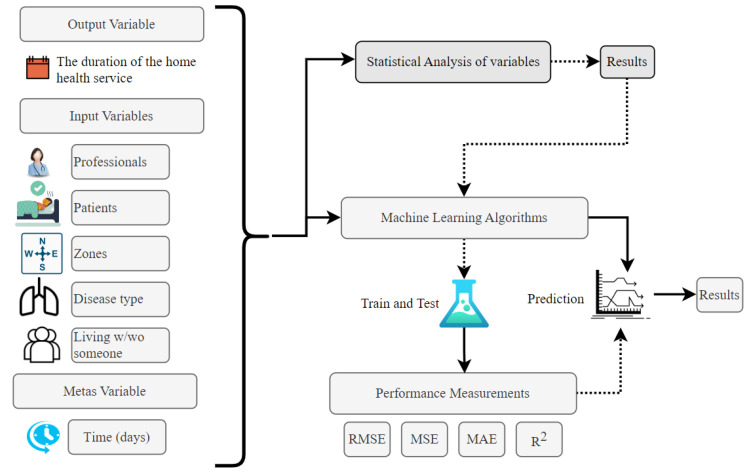
The flowchart of the method proposed.

**Figure 3 healthcare-11-00319-f003:**
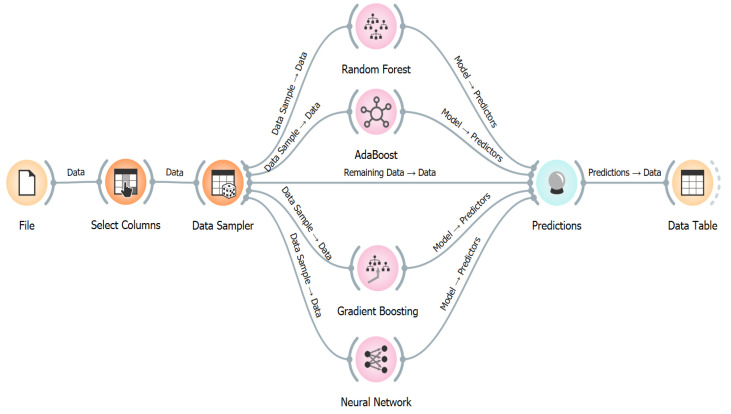
The software screenshot of AB, RF, NN, and GB models based on the ML.

**Figure 4 healthcare-11-00319-f004:**
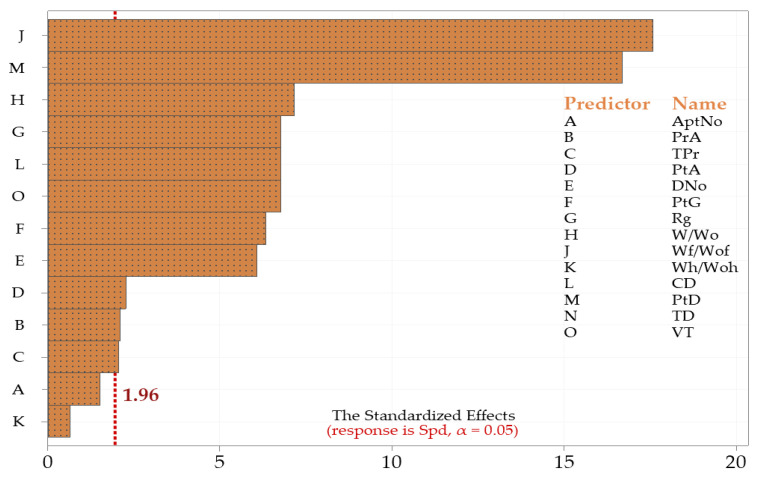
The standardized effect graph of the dependent variable.

**Figure 5 healthcare-11-00319-f005:**
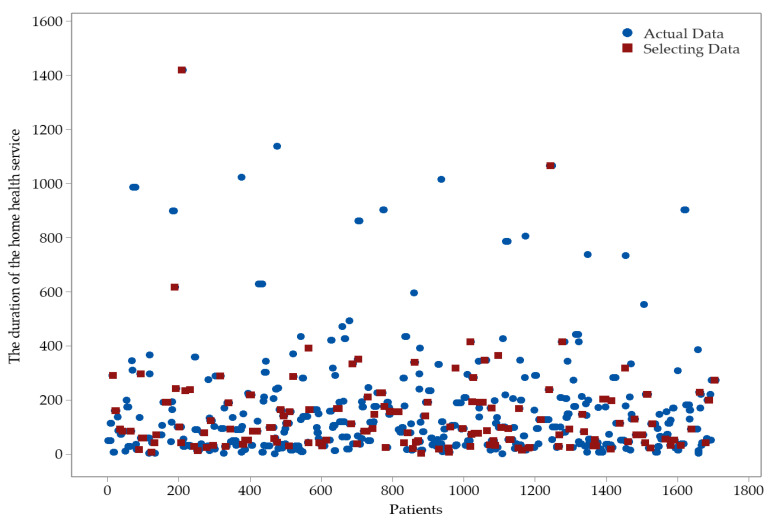
The data selected for the testing phase in the dataset.

**Figure 6 healthcare-11-00319-f006:**
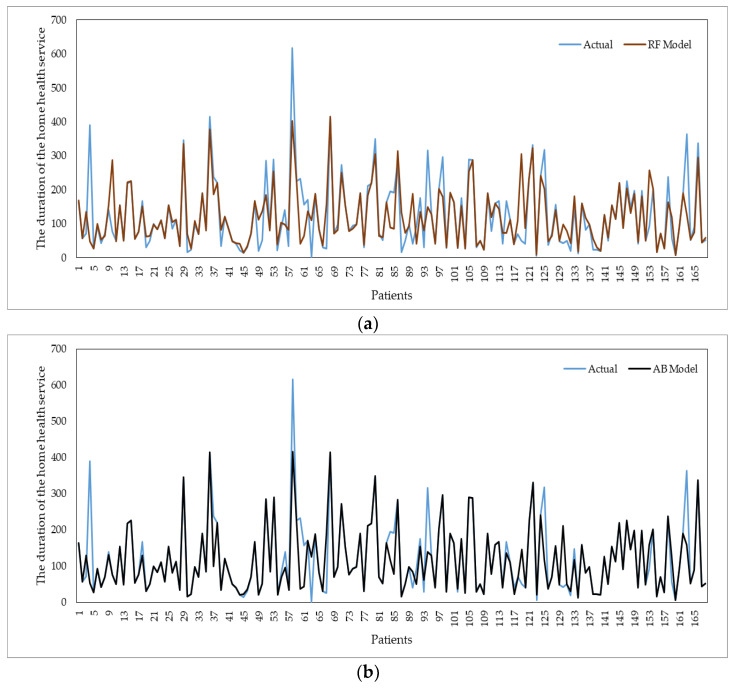
Comparison of forecast data of ML algorithms with real data (**a**) actual value vs. RF model, (**b**) actual value vs. AB model, (**c**) actual value vs. GB model, (**d**) actual value vs. NN model.

**Figure 7 healthcare-11-00319-f007:**
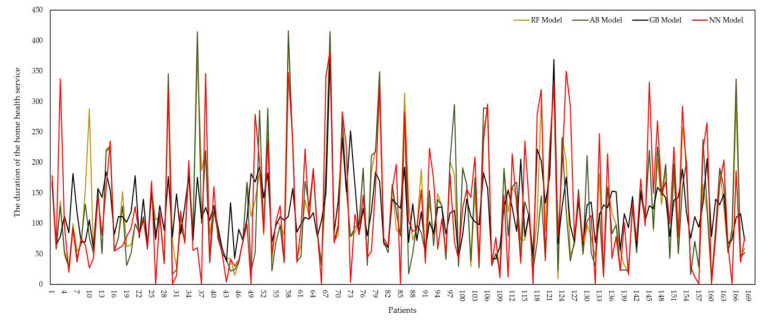
Prediction data of RF, AB, GB, and NN algorithms.

**Table 1 healthcare-11-00319-t001:** Definition, type, notation, and data type information of dependent and independent variables.

Definition of Variables	Types of Variables	Notation of Variables	Types of Data
The door number of the house/apartment where the patient lives (for floor info)	Input	AptNo	Numeric
Personal Age	Input	PrA	Numeric
Work experience period of the personnel responsible for the home health practice (Tenure)	Input	TPr	Numeric
Patient Age	Input	PtA	Numeric
Districts where the patient lives	Input	Dno	Numeric
Patient Gender	Input	PtG	Categoric
Regions responsible for the districts	Input	Rg	Categoric
Does the patient live alone at home?	Input	W/Wo	Categoric
Does anyone stay with the patient all day long?	Input	Wf/Wof	Categoric
Is there someone in the family or nearby to help?	Input	Wh/Woh	Categoric
About whether the patient has any or more chronic diseases *	Input	CD	Categoric
Does the patient have a disability?	Input	PtD	Categoric
Types of Disability	Input	TD	Categoric
Visiting Types	Input	VT	Categoric
Service Period (Day)	Output	SPd	Numeric
Service Period (Year)	Output	SPy	Numeric

* The chronic disease factor, one of the independent factors of this study, includes approximately 70 types of chronic diseases. The list of these diseases is shared in Table A1 in Appendix A the study.

**Table 2 healthcare-11-00319-t002:** The values of descriptive statistics of independent and dependent variables based on the region factor.

Variable	Rg	N	Mean	SE Mean	StDev	Min	Max	Skewness	Kurtosis
AptNo	Edirnekapi	402	4.54	0.21	4.28	0.00	21.00	1.96	4.11
	Kartal	527	7.67	0.42	9.55	0.00	43.00	2.48	6.13
	Kucukcekmece	200	7.80	0.94	13.32	1.00	90.00	3.94	17.22
	Uskudar	578	7.03	0.38	9.02	1.00	145.0	7.04	94.88
PrA	Edirnekapi	402	37.09	0.42	8.49	23.00	52.00	0.00	−1.16
	Kartal	527	36.80	0.46	10.39	23.00	56.00	0.12	−1.28
	Kucukcekmece	200	39.18	0.50	7.11	27.00	49.00	−0.50	−1.28
	Uskudar	578	35.47	0.35	8.38	23.00	58.00	0.34	−1.19
TPr	Edirnekapi	402	8.43	0.27	5.32	0.66	18.66	0.03	−1.14
	Kartal	527	6.58	0.18	3.99	1.75	13.73	0.38	−1.25
	Kuçukcekmece	200	9.66	0.36	5.15	0.62	18.07	−0.20	−1.15
	Uskudar	578	7.05	0.20	4.79	0.62	16.53	0.23	−1.09
PtA	Edirnekapi	402	70.42	0.79	15.86	18.00	97.00	−0.59	0.30
	Kartal	527	72.57	0.70	15.99	17.00	100.0	−0.94	0.62
	Kucukcekmece	200	66.99	1.39	19.66	9.00	97.00	−0.85	0.66
	Uskudar	578	71.53	0.60	14.32	18.00	93.00	−0.82	0.06
DNo	Edirnekapi	402	19.86	0.36	7.29	8.00	37.00	0.14	−0.34
	Kartal	527	27.45	0.12	2.73	24.00	34.00	1.01	0.53
	Kucukcekmece	200	14.49	0.59	8.39	1.00	30.00	0.05	−1.41
	Uskudar	578	24.86	0.52	12.40	2.00	36.00	−0.61	−1.15
Spd	Edirnekapi	402	137.6	6.27	125.74	3.00	861.0	2.72	11.53
	Kartal	527	189.9	10.90	250.30	2.00	1420	2.64	7.44
	Kucukcekmece	200	139.7	9.26	130.99	1.00	1015	2.26	9.31
	Uskudar	578	101.8	5.80	139.37	1.00	1023	4.45	24.46
Spy	Edirnekapi	402	0.38	0.02	0.34	0.01	2.36	2.72	11.53
	Kartal	527	0.52	0.03	0.69	0.01	3.89	2.64	7.44
	Kucukcekmece	200	0.38	0.03	0.36	0.00	2.78	2.26	9.31
	Uskudar	578	0.28	0.02	0.38	0.00	2.80	4.45	24.46

Abbreviation: N, the total count of the sample set; SE, Standard error; StDev, standard deviation; Min, the minimum value; Max, the maximum value.

**Table 3 healthcare-11-00319-t003:** Statistical values of the effects of independent variables on the duration of the home health service.

Source	DF *	Adj SS	Adj MS	F-Value	*p*-Value
Regression	212.0	46,844,959.0	220,967.0	94.56	0.001
AptNo	1.000	5384.00000	5384.000	2.300	0.129
PrA	1.000	10,265.0000	10,265.00	4.390	0.036
TPr	1.000	9842.00000	9842.000	4.210	0.040
PtA	1.000	11,944.0000	11,944.00	5.110	0.024
DNo	1.000	85,886.0000	85,886.00	36.75	0.001
PtG	1.000	93,915.0000	93,915.00	40.19	0.001
Rg	3.000	2,400,822.00	800,274.0	342.5	0.001
W/Wo	1.000	119,777.000	119,777.0	51.26	0.001
Wf/Wof	1.000	721,694.000	721,694.0	308.9	0.001
Wh/Woh	1.000	938.000000	938.0000	0.400	0.526
CD	194.0	37,557,001.0	193,593.0	82.85	0.001
PtD	1.000	651,308.000	651,308.0	278.7	0.001
VT	5.000	212,863.000	42,573.00	18.22	0.001

Abbreviation: DF, degree of freedom; Adj SS; the adjusted sum of squares; Adj MS, adjusted means of squares. * The sum of the degrees of freedom of the independent variables is 424, and the number of degrees of freedom of the remaining data is 824 (lack-of-fit) and 649 (Pure Error). The total number of degrees of freedom used is 1897.

**Table 4 healthcare-11-00319-t004:** Performance measurement values of testing and training phases.

Phases	Models	MSE	RMSE	MAE	R^2^
Training	RF	0.015	0.122	0.045	0.941
	AB	0.000	0.004	0.000	0.990
	GB	0.087	0.296	0.189	0.654
	NN	0.038	0.195	0.101	0.849
Test	RF	0.029	0.170	0.080	0.848
	AB	0.018	0.136	0.043	0.903
	GB	0.051	0.225	0.175	0.733
	NN	0.091	0.302	0.164	0.521

**Table 5 healthcare-11-00319-t005:** Performance measurement values of ML algorithms.

Models	MSE	RMSE	MAE	R^2^
RF	0.026	0.163	0.078	0.861
AB	0.018	0.136	0.043	0.903
GB	0.051	0.225	0.175	0.733
NN	0.091	0.302	0.164	0.521

## Data Availability

The study did not report any data.

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
