# Peer review of "Analysis of Home Healthcare Practice to Improve Service Quality: Case Study of Megacity Istanbul"

_healthcare, 2023, doi:10.3390/healthcare11030319_

Round 1
Reviewer 1 Report
This study provides a way to analyze the factors affecting the increased quality of home healthcare practices. This study offers exciting research by addressing a different subject. The work sections have been designed appropriately, and these sections have been handled well by the authors.
Comments and Suggestions for Authors:
Line 44-48: This sentence in the study should be supported by a reference.
For data in the first column of all tables: Information in tables should be left or right-aligned. Usually left-aligned used.
Explanations expressing the connection between LP and ML algorithms should be added.
Some grammatical/typing errors should be corrected.
The conclusion part needs to be promoted with the results of some studies. The conclusion should be apt and dwell on critical points readers can pick from the study. In addition, policy implications should be from the findings, not general. A research paper's essence is to develop policy suggestions for policymakers, government, firms, or interested individuals. This aspect is neglected and thus makes the whole effort futile. Please revisit and expand this section.
Author Response
Respected Reviewer,
We would like to thank you for reviewing our article. Criticisms and comments are, of course, very valuable in correcting our shortcomings and mistakes. We appreciate your valuable comments, contributions, and suggestions. We have made the necessary arrangements within the framework of your suggestions and questions and according to the recommendations of the other reviewers.
All changes in letters, words, and sentences were painted with red color in the manuscript.
Thank you for your consideration of our article. We hope that with our modifications, we have met your expectations. This letter is a point-by-point response to the comments.
The revision report is attached.
Kind Regards,
The Corresponding Author

Reviewer 2 Report
This study aimed to increase the quality of home health services by examining the services provided to socially disadvantaged, sick, needy, disabled, and elderly individuals and by analyzing the factors affecting home health practices. In this study, 14 input variables affecting home healthcare practices in Istanbul were considered using big data. The main focus of this study's input parameters is on the patients and healthcare workers who work in home health practices and their age, location, and living situation. This is how the data was organized.
Overall, this is a well-written paper with clear explanations of how it was done and what it found. The use of routine nationwide data is interesting, and advanced/sophisticated statistical procedures were applied. I have a few suggestions the authors might want to consider
The first sentence of the second paragraph of the introductory part of the study should be more understandable.
In the study, some words were expressed as 'healthcare' or 'health care'. These words should be written as a single type.
The conclusion part of the study should be expanded.
Page 4, lines 177–178: The day and year definitions of the dependent variable's data set should be expanded.
The author should correct grammar and spelling errors.
Author Response

(The authors gave the same response as above.)
